# Evidence for two dimensional anisotropic Luttinger liquids at millikelvin temperatures

Guo Yu [1,2,6], Pengjie Wang [1,6], Ayelet J. Uzan-Narovlansky [1], Yanyu Jia [1], Michael Onyszczak [1], Ratnadwip Singha [3], Xin Gui [3], Tiancheng Song [1], Yue Tang [1], Kenji Watanabe [4], Takashi Taniguchi [5], Robert J. Cava [3], Leslie M. Schoop [3] & Sanfeng Wu [1] ✉

Interacting electrons in one dimension (1D) are governed by the Luttinger liquid (LL) theory in which excitations are fractionalized. Can a LL-like state emerge in a 2D system as a stable zero-temperature phase? This question is crucial in the study of non-Fermi liquids. A recent experiment identified twisted bilayer tungsten ditelluride (tWTe₂) as a 2D host of LL-like physics at a few kelvins. Here we report evidence for a 2D anisotropic LL state down to 50 mK, spontaneously formed in tWTe₂ with a twist angle of ~ 3°. While the system is metallic-like and nearly isotropic above 2 K, a dramatically enhanced electronic anisotropy develops in the millikelvin regime. In the anisotropic phase, we observe characteristics of a 2D LL phase including a power-law across-wire conductance and a zero-bias dip in the along-wire differential resistance. Our results represent a step forward in the search for stable LL physics beyond 1D.

The Luttinger liquid (LL) theory of 1D interacting conductors offers a demonstration of a gapless electronic phase beyond the standard Fermi liquid (FL) paradigm[1,2]. Distinct features of a LL owing to strong correlations include the power-law suppression of the density of state (DOS) at the Fermi energy and the fractionalization of electronic excitations into collective modes associated separately with spin and charge degrees of freedom. These appealing properties of a 1D LL led Anderson to explore the possibility of LL-like physics in dimensions higher than one for explaining the unusual phenomena of cuprate superconductors[3–6]. Theoretical searches for 2D or 3D LL-like non-Fermi liquids were put forward in the context of coupled-wire constructions[7–11], where identical parallel 1D nanowires, each being described by the LL theory, are placed together to form 2D arrays or 3D networks.

One key question is whether the LL physics survives in the coupled-wire systems at vanishing temperatures when interwire hopping is turned on. Similar problems have been considered in the study of quasi-1D organic conductors[12], which may be regarded as weakly coupled wires. Typically, a crossover temperature $T^*$ from the LL state at intermediate temperatures to a 2D FL or gapped state at low temperatures exists and may be estimated[1,7,13] as $T^* \sim t_\perp (t_\perp/t_\parallel)^{\eta/(1-\eta)}$, where $t_\perp$ ($t_\parallel$) is the interwire (intrawire) hopping term and $\eta$ is the power law exponent of the DOS at the 1D Fermi surface[13]. Here $\eta$ reflects the intrawire interaction strength (see illustration in Fig. 1a, b) and $t_\perp < t_\parallel$ in a coupled wire setting. Experimentally, $T^*$ in organic quasi-1D conductors is typically tens of kelvins[12], below which the LL description is invalid. Interestingly, in the above expression, one readily sees that the intrawire interaction suppresses $T^*$. If $\eta$ is large enough, i.e., $\eta > 1$, $T^*$ in principle vanishes, indicating a new regime where single-particle hopping is irrelevant[1,7,13] even down to zero temperature. In realistic systems, other competing phases, especially those arising from two-particle hopping processes, become important in this regime and provide instabilities to the LL state. Evaluating competing phases, such as charge density wave, FL and superconducting states, is indeed the key focus of multiple theoretical works[7–11] in the early 2000s, which carefully investigated the phase diagram of a 2D array of coupled LL.

[1]Department of Physics, Princeton University, Princeton, NJ 08544, USA. [2]Department of Electrical and Computer Engineering, Princeton University, Princeton, NJ 08544, USA. [3]Department of Chemistry, Princeton University, Princeton, NJ 08544, USA. [4]Research Center for Functional Materials, National Institute for Materials Science, 1-1 Namiki, Tsukuba 305-0044, Japan. [5]International Center for Materials Nanoarchitectonics, National Institute for Materials Science, 1-1 Namiki, Tsukuba 305-0044, Japan. [6]These authors contributed equally: Guo Yu, Pengjie Wang. ✉e-mail: sanfengw@princeton.edu

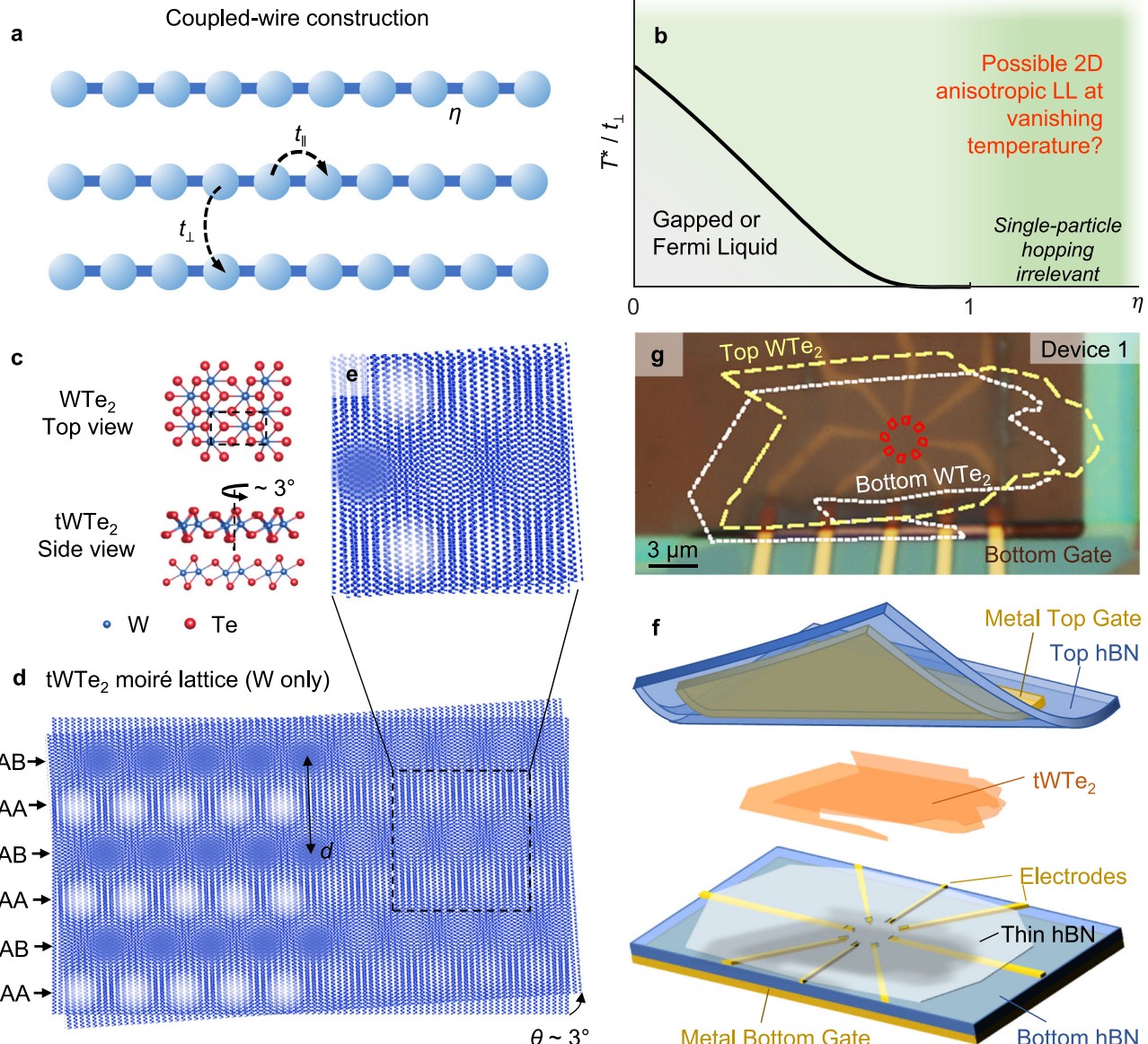

**Fig. 1 | The coupled-wire construction and tWTe₂ moiré superlattices. a** Cartoon illustration of coupled-wire construction, where the interwire (intrawire) hopping $t_\perp$ ($t_\parallel$) and the effective power law exponent $\eta$ in a single wire is indicated. **b** A sketch of the 1D-2D crossover temperature $T^*$ *v.s.* $\eta$ in coupled wires, only considering single particle hopping. **c** Top, a top view of monolayer WTe₂, with the unit cell marked by the dashed rectangle. Bottom, a side view of a tWTe₂ with the top layer rotated by 3°. **d** Moiré superlattice of tWTe₂ (Only W atoms for a better visualization), showing alternating AB and AA stacking sites. The AA (or AB) sites mimic the coupled-wire construction shown in (**a**). **e** A zoom-in view of the tWTe₂ moiré lattice. **f** A cartoon illustration of the device structure. See Methods and Supplementary Fig. 1 for details about device fabrication. **g** An optical image of device 1. Yellow dashed line, white dotted line and red solid line indicate respectively the top WTe₂, bottom WTe₂, and the electric contact regions.

Under some fine-tuned interactions, they predicted, remarkably, the possible existence of a stable anisotropic phase in which both single- and two-particle interwire hopping processes are irrelevant, i.e., a phase akin to the LL surviving as a zero-temperature ground state in a small parameter space. This anisotropic 2D phase, coined a "sliding LL"[9–11] or "smectic metal"[8], is surrounded by its competing orders and its realization requires careful control of parameters in the coupled wires.

An experimental realization of coupled-wire constructions in a controllable setting is however challenging. Very recently, new material systems have been identified for investigating LL physics beyond 1D, notably including the moiré superlattice of twisted bilayer WTe₂ (tWTe₂)[14] and a quasi-2D material η-Mo₄O₁₁[15]. Evidence for LL physics has been shown down to 1.8 K in tWTe₂ with a twist angle near 5° and ~10 K in η-Mo₄O₁₁. An essential question is whether the LL state can

survive in any realistic systems down to the lowest achievable temperature in experiments, particularly in the millikelvin regime. In this work, we address this fundamental question for establishing the concept of a 2D LL, based on tWTe₂ moiré superlattices. The exceptional tunability of electronic properties by moiré engineering places tWTe₂ as an outstanding system for such studies.

## Results

### tWTe₂ Moiré lattice and device design

Owing to the rectangular unit cell of monolayer WTe₂ (Fig. 1c), the super unit cell of the moiré lattice in tWTe₂, when the twisted angle $\theta$ is small, is also a rectangle but large. Figure 1d, e illustrate the atomic structure of tWTe₂, where only the W atoms are shown to better visualize it. The moiré pattern clearly resembles the coupled-wire lattice shown in Fig. 1a. By tuning the twist angle, one can arbitrarily

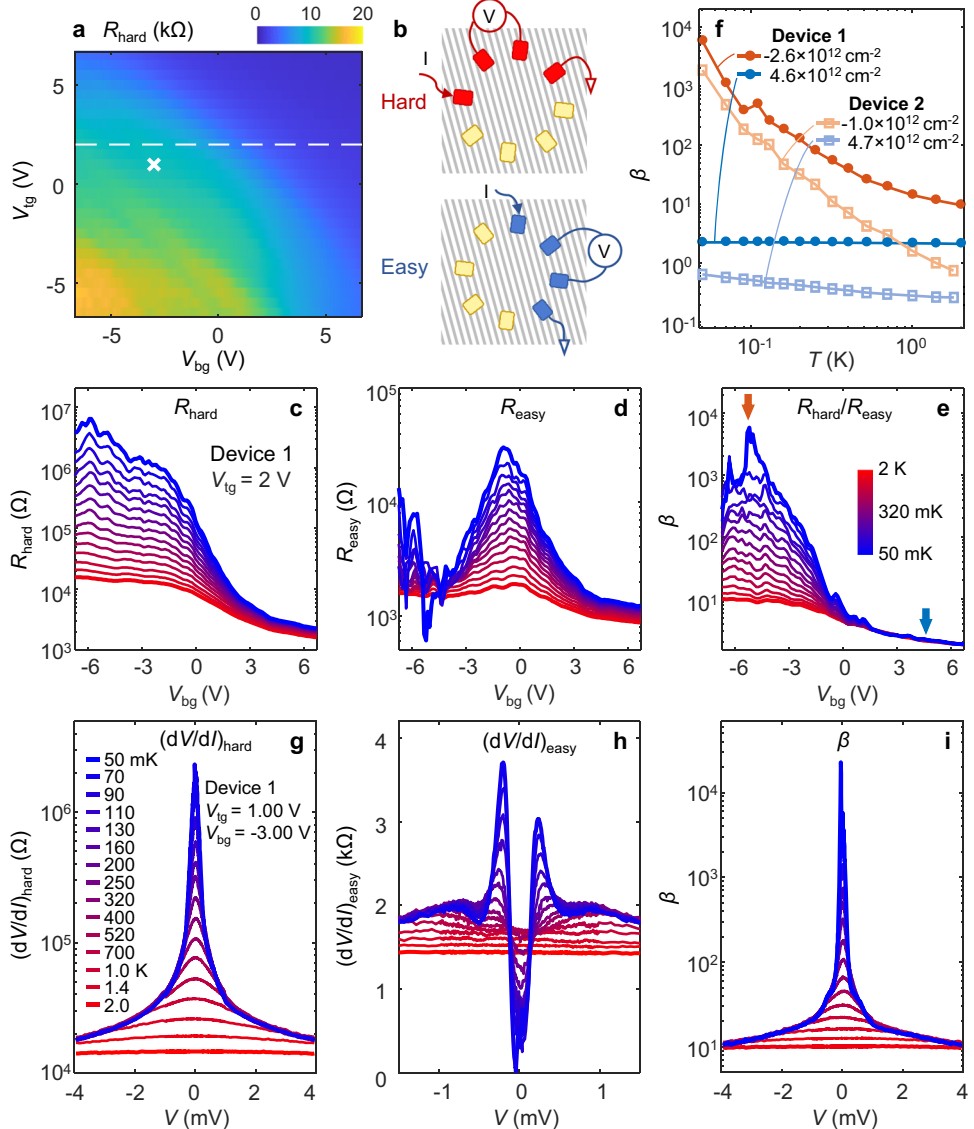

**Fig. 2 | Exceptional transport anisotropy at millikelvin temperatures. a** A dual-gate map of resistance ($R_{hard}$, geometry shown in **b**) taken in device 1 at 4 K. **b** Cartoon illustration of measurement geometries for $R_{hard}$ and $R_{easy}$. **c** $R_{hard}$ *v.s.* $V_{bg}$ taken at various $T$, ranging from 50 mK (blue) to 2 K (red). See inset in (**e**–**g**) for temperature legends. $V_{tg}$ is kept at 2 V as indicated by the dashed line in (**a**). **d** Same measurements as in (**c**) but for $R_{easy}$. **e** The gate-dependent anisotropy ratio $\beta \equiv$ $R_{hard}/R_{easy}$ under various $T$. Orange (blue) arrow indicates the gate where the orange (blue) curve in (**f**) is extracted. **f** $\beta$ as a function of $T$, plotted for two typical $n_g$ in electron (hole) side. Data taken in both devices 1 and 2 are shown. **g** ($dV/dI$)$_{hard}$ *v.s.* d.c. bias $V$ at various $T$ at a selected gate parameter indicated by the cross in (**a**). **h** The same as (**g**) but for easy direction. **i** Bias-dependent anisotropy $\beta$ at various $T$.

choose the size of the supercell in a wide range, which alters key parameters that determine its ground phases. Our previous work revealed that tWTe$_2$ at $\theta \sim 5°$ indeed develops LL physics below ~30 K on the hole-rich side[14], evidenced by the emergence of large transport anisotropy and power law scaling behaviors of its conductance. There the LL transport characteristics were confirmed down to 1.8 K, below which the across-wire resistance becomes too large (>10 MΩ) to be resolved quantitatively[14], preventing transport access to the physics in the sub-kelvin regime. The essential question of whether the LL description is valid at millikelvin temperatures in tWTe$_2$, or any 2D/3D experimental system, remains unknown.

In this work we focus on tWTe$_2$ with a smaller $\theta$, near 3°, where the moiré cell is larger (interwire distance $d \sim 12$ nm, as shown in Fig. 1d) and the energy scale is in principle smaller. Similar to previous reports[14,16,17], we fabricate devices with tWTe$_2$ fully encapsulated by top and bottom hexagonal boron nitride (hBN) dielectrics and metal (Pd) gate (see Fig. 1f for device structure). A thin layer of selectively etched hBN is inserted

between the tWTe$_2$ and metal (Au or Pd) electrodes, to ensure direct electric contacts to the tWTe$_2$ interior (indicated by the red squares in the optical image of a typical device shown in Fig. 1g). With this device geometry, contributions to transport from the nearby monolayer WTe$_2$ regions, as well as its edge modes, are minimized. Details of device fabrication are illustrated in Methods and Supplementary Fig. 1. The application of gate voltages varies the carrier density in the sample. We quantify the gate-induced doping as $n_g \equiv \varepsilon_r \varepsilon_0 (V_{tg}/d_{tg} + V_{bg}/d_{bg})/e$, where $V_{tg}$ ($V_{bg}$) is the top (bottom) gate voltage; $d_{tg}$ ($d_{bg}$) is the thickness of the top (bottom) hBN dielectric; $\varepsilon_0$, $\varepsilon_r$ and $e$ are respectively the vacuum permittivity, relative dielectric constant of hBN, and elementary charge. The choice of near 3° twist angle is based on systematic studies of tWTe$_2$ with a range of small twist angles (see Supplementary Figs. 2–4). The much smaller resistivity, compared to ~5° devices or the monolayer, together with a large emergent anisotropy near this angle, provides a key condition for us to evaluate the LL transport characteristics quantitatively down to temperatures as low as ~50 mK.

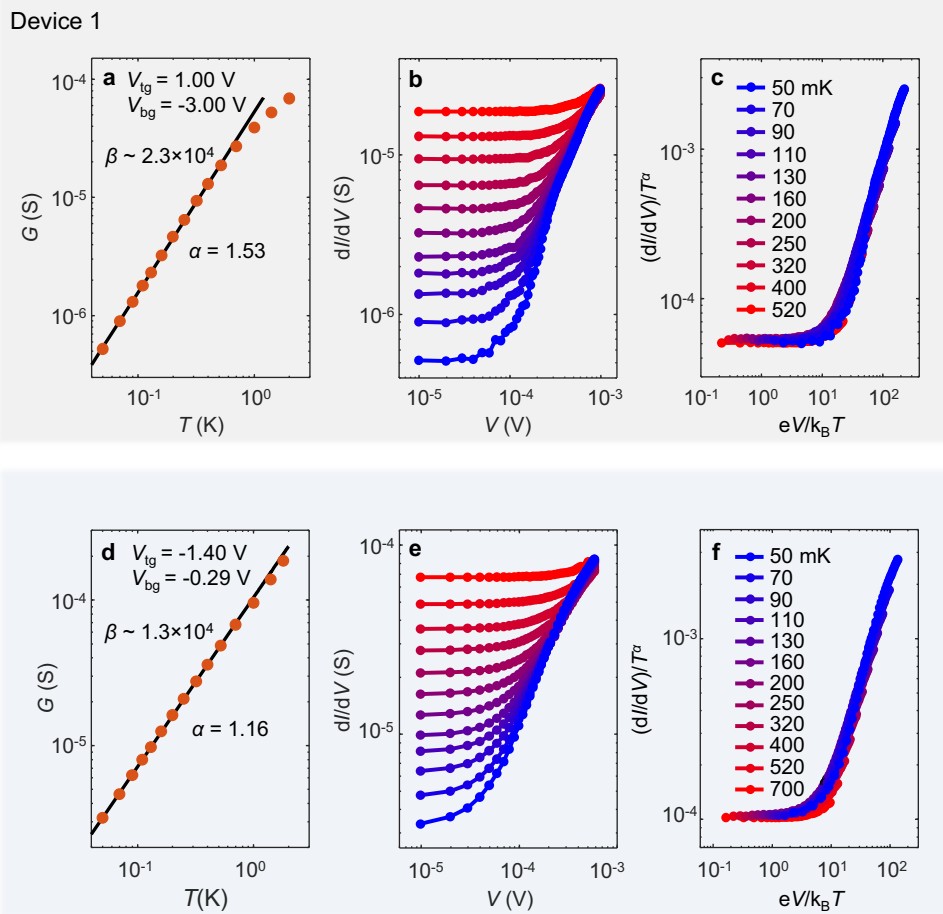

**Fig. 3 | Luttinger Liquid behaviors down to 50 mK. a** $G$ $v.s.$ $T$ in a log-log plot taken in device 1 at the same gate configuration as that in Fig. 2g–i. The contact configuration is the same as $R_{hard}$. The solid line is a power law fit to the low $T$ data, with the resulting exponent α indicated. The corresponding anisotropy $β$ is indicated as well. **b** d$I$/d$V$ $v.s.$ $V$ at various $T$, ranging from 50 mK to 520 mK (the same legends as (**c**)), taken at the same gate voltages as in (**a**). **c** The scaled plot (d$I$/d$V$)/$T^α$ $v.s.$ $eV/k_B T$ for the same data shown in (**b**) using the same α extracted in (**a**). **d–f** The same plots for a selected gate configuration taken in device 2.

## Emergent anisotropy at millikelvin temperatures

Figure 2a plots resistance measured in device 1 ($θ$ ~ 3°) under varying gate voltages at a sample temperature, $T$, of 4 K. The maximum is only ~20 kΩ and no substantial transport anisotropy is found; both aspects are distinct from the tWTe$_2$ with $θ$ ~ 5° at the same temperature[14], highlighting the key role of $θ$. Interestingly, an exceptionally large transport anisotropy develops at millikelvin temperatures, where the easy and hard transport directions can be clearly identified by measuring resistances between neighboring probes in the ring contact geometry (see Supplementary Fig. 5). To quantify the anisotropy, we carefully examine four-probe resistances ($R_{easy}$ and $R_{hard}$) measured in both directions (as illustrated in Fig. 2b). Figure 2c, d plot $R_{hard}$ and $R_{easy}$, respectively, as a function of $V_{bg}$ at a fixed $V_{tg}$ = 2.00 V, taken at various $T$ below 2 K. One clearly sees a dramatic increase of $R_{hard}$ on the hole-rich side and near the charge neutrality point (CNP) when $T$ is lowered. At 50 mK, $R_{hard}$ reaches a value of >1 MΩ. In sharp contrast, this strong increase is absent on the electron-rich side and, markedly, for $R_{easy}$ at all doping. We define an anisotropy ratio $β ≡ R_{hard} / R_{easy}$ (Fig. 2e). Near CNP and with hole doping, $β$ is large, reaching 10,000 at 50 mK. Warming the sample $β$ decreases dramatically (Fig. 2f). Anisotropy is absent in the electron-rich regime at any $T$.

The distinct transport along the two orthogonal directions (easy $v.s.$ hard) manifests itself not only in the strong anisotropy (large $β$), but also in its bias-dependence. Figure 2g–i examine the effects of a d.c. bias ($V$) applied to the source contact in device 1, at $V_{tg}$ = 1.00 V and

$V_{bg}$ = -3.00 V (indicated by the white cross in Fig. 2a), where a large $β$ is seen. Near zero bias, a large peak is clearly seen when transport is along the hard direction (Fig. 2g), exhibiting an insulating-like behavior. Warming up the device to ~2 K, the curve flattens. In sharp contrast, the same measurement along the easy direction yields a clear zero bias dip (Fig. 2h). Similar behavior is found in device 2 ($θ$ ~ 3.5°), as shown in Supplementary Fig. 6. A consistent zero bias dip is also seen in a ~ 5° tWTe$_2$ device (Supplementary Fig. 7). At first glance, this dip feature resembles that of a superconductor. However, it cannot be suppressed by magnetic fields (Supplementary Fig. 6) and only appears when it is measured along the easy direction. Instead of arising from superconductivity, this remarkable feature can be well explained by the spontaneous formation of a new phase consisting of a 2D array of 1D electronic channels, as illustrated by the gray lines in Fig. 2b. At small bias, this new phase develops and the current flowing between source and drain contacts are restricted to the 1D channels connecting them. Consequently, current flow is minimized between the voltage probes placed nearby, yielding a vanishing voltage, i.e., the zero-bias dip. At high $V$ or high $T$, this strongly anisotropic phase is destroyed (Fig. 2i), leading to a finite voltage between the two probes. The shoulder next to the zero-bias dip (Fig. 2h) signifies the transition between the anisotropic phase to an isotropic one at high $V$.

## LL Characteristics down to 50 mK

We next examine transport characteristics expected for a LL state, namely the power low scaling behavior of the conductance.

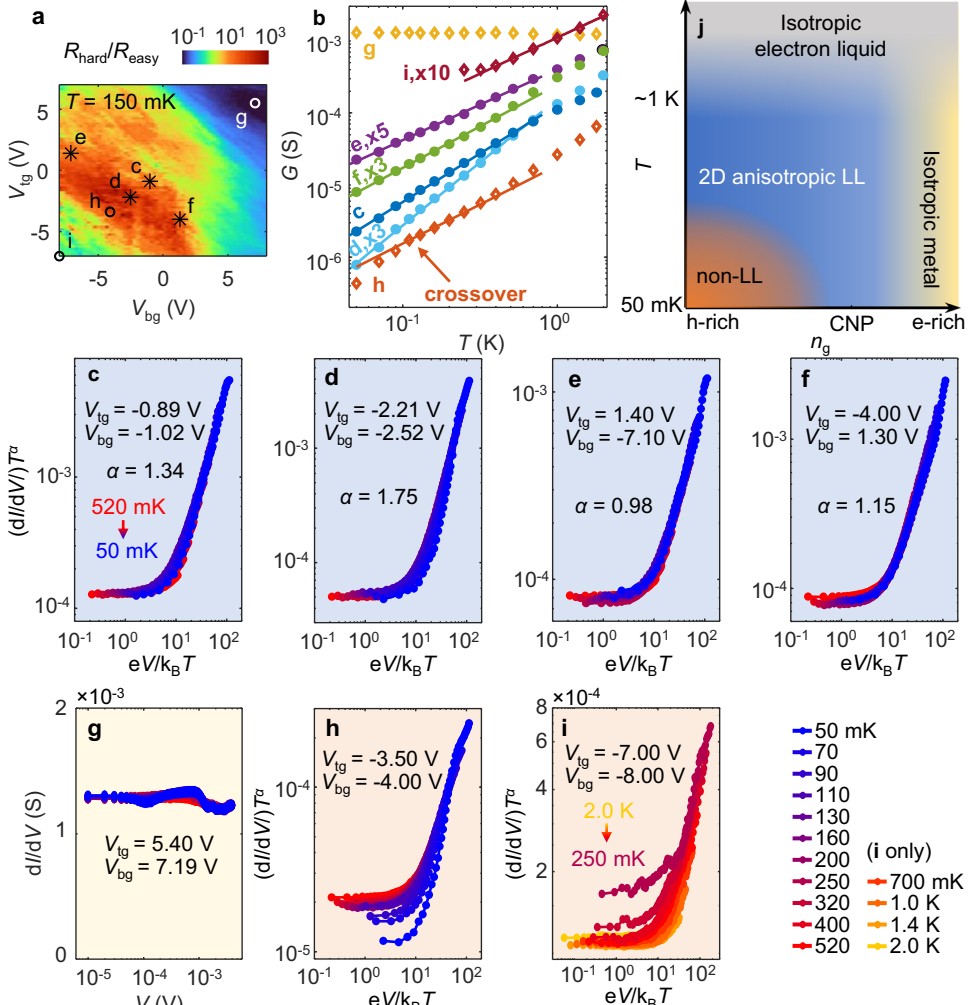

**Fig. 4 | Electronic phase diagram for the tWTe₂ ($\theta \sim 3°$). a** A dual gate map of anisotropy $\beta$ taken at 150 mK (device 2). **b** $G$ $v.s.$ $T$ at selected gate voltages, corresponding to the spots marked as (**c–i**) in (**a**). For better visualization, some curves are multiplied by a factor, as indicated next to them. **c–f** Scaled differential conductance plot for the corresponding spots (**c–f**) in (**a**). Excellent power law scaling behaviors are seen, indicating the LL physics. **g** $dI/dV$ $v.s.$ $V$ for spot g (electron-rich region), showing a weak $T$ or $V$ dependence (an Ohmic behavior). **h** The scaled differential conductance plot for spot h, which develops a clear deviation from the universal scaling. An exponent $\alpha = 1.08$ is used in the plot. **i** The same scaling plot for spot i, which also develops a deviation from the universal scaling. Data is present down to 250 mK, below which electric contacts become bad at this gate configuration. An exponent $\alpha = 1.00$ is used in the plot. **j** A preliminary phase diagram for the -3° tWTe₂ system.

Conventionally, in a single 1D wire system one may measure tunneling conductance from a Fermi liquid lead to the wire[18–22]. In our case of an array of parallel 1D wires in the moiré system, electron transport in the hard direction involves tunneling between wires, providing an excellent opportunity for examining power law behaviors without the need of an external tunneling probe[14]. One consequence of the LL physics is that the across-wire conductance $G(T) \equiv 1/R_{hard} \propto T^\alpha$ where the exponent $\alpha$ reflects the power law suppression of the DOS near the Fermi energy[7–11,13,23]. This is indeed seen in the strongly anisotropic regime of our devices, as shown in Fig. 3a for device 1. At this gate configuration, we find that a value of $\alpha \sim 1.53$ captures well the low-$T$ conductance from ~500 mK down to 50 mK. A transition to the high-$T$ isotropic phase occurs near ~1 K, above which only a weak $T$-dependence is seen in $G$. In Supplementary Fig. 8, we further confirm that neither an exponential form expected for an activation gap nor the variable-range hoping form for localization[24] can describe the observed conductance at millikelvin temperatures.

Another essential LL feature lies in the bias-dependent differential conductance, i.e., $dI/dV \propto V^\alpha$ when $eV \gg k_B T$, where $k_B$ is the Boltzmann constant. The same exponent $\alpha$ must be seen here as in the above $G(T)$

since it reflects the same suppression of DOS. In other words, LL physics[1] dictates that the scaled conductance $(dI/dV)/T^\alpha$ is only a function of $eV/k_B T$. Figure 3b plots the measured $dI/dV$ as a function of $V$ varied from 10 μV to ~1 mV, taken at various $T$. Remarkably, all data points, taken in the parameter space spanned over two decades in $V$ and one decade in $T$, collapse into a single curve in the scaled plot (Fig. 3c)! The only parameter used here is $\alpha$, the same one extracted from $G(T)$ (Fig. 3a). Hence the exponent $\alpha$ provides a key description of the transport behaviors of the system, well consistent with the emergence of LL physics in the moiré system. In a simplified argument where only an effective intrawire Fermi surface exponent $\eta$ is considered, a calculation for across wire transport yields $\alpha = 2\eta - 1$[13]. If this is valid, we estimate $\eta \sim 1.26$, a value that is larger than one. It is therefore consistent with the condition for the single-particle interwire hopping to be irrelevant, as discussed in Fig. 1b. We note that in realistic system, two-particle hopping process are also important in this regime and hence the interactions shall be described by more complex parameters that involve both interwire and intrawire interactions[7–11], instead of a single $\eta$. In that case, dimensional crossover and competing orders provide new instabilities to the LL phase. Theoretically,

competing phases indeed reside in most regions in the calculated phase diagram, expect that the predicted sliding LL under fine-tuned interactions offers a possible realization of a 2D LL phase at vanishing tempeatures[7–11]. The experimental consequence is that although the wires are closely packed, the system, driven by interactions, behaves as an array of "independent" LL wires, and that transport across the wires is fully suppressed unless a finite temperature or bias is applied. From this perspective, the vanishing d$V$/d$I$ (Fig. 2h) in the zero-bias dip measured along the wires, which does indicate that the wires are effectively independent, provides an additional key characterization of the observed phase in tWTe$_2$. We believe our observations of the dramatic zero-bias dip along the wire, together with the large anisotropy and the power law across-wire conductance, indicate the emergence of a highly intriguing new phase akin to the proposed "sliding LL", although the understanding of the exact mechanism requires substantial future developments in both theory and experiments.

The observations are reproduced in different contact geometries (Supplementary Fig. 9) and in device 2, as shown in Fig. 3d–f and Supplementary Fig. 6. We again highlight that here the LL description is valid down to 50 mK, an unprecedented regime. This temperature is well below the energy scale (~meV) of the hopping and interaction terms in the system. Any dimensional crossover, if exists, must be lower than this temperature.

### Electronic phase diagram

We discuss the gate-tuned phase diagram of tWTe$_2$ at this small twist angle based on device 2. Figure 4a presents the gate-dependent anisotropy map taken at 150 mK, where the red color indicates large $\beta$. $G(T)$ and d$I$/d$V$ were recorded at selected typical locations, as indicated in the map. Power law behaviors are found together with strong anisotropy for locations labeled as c-f, while regions of g, h and i show clear deviations from a power law (Fig. 4b). Particularly, at c-f, electronic transport exhibits universal scaling characteristics (Fig. 4c–f) qualitatively like the observations in Fig. 3, hence implying the formation of a 2D anisotropic LL phase robust down to 50 mK. This region occurs near CNP, with $n_g$ roughly within $\pm 4 \times 10^{12}$ cm$^{-2}$. We find that while LL behaviors are sensitive to carrier density, the displacement field ($D$) effect is less dramatic especially if $D$ is not too large. At high $D$, we do find a drop in the power law exponent (Supplementary Fig. 10), potentially indicating a transition to a different phase if $D$ is further increased. On the electron dominant side (e.g., location g), one sees almost no $T$- or $V$- dependence of the conductance, indicating a transition to a metallic-like state with an Ohmic behavior (Fig. 4g). On the hole dominant side, although strong transport anisotropy starts to develop below ~1 K, the conductance deviates from the power law typically when $T$ is further lowered to, e.g., ~400 mK for location i and ~100 mK for h, as shown in Fig. 4b, h & i. The data suggests a crossover to a non-LL phase at lower $T$. In Fig. 4j we summarize the observation by presenting a preliminary phase diagram describing tWTe$_2$ at this twist angle, under varying $T$ and $n_g$ (here we limit the electric displacement field to small values for simplicity). The presence of a finite parameter region (blue) that hosts an anisotropic 2D phase mimicking the LL is the key finding.

### Discussion

In this work, we experimentally address the long-standing question of whether a 2D non-Fermi liquid phase resembling a LL can exist as a stable ground state. We conclude that such a phase does develop in the tWTe$_2$ moiré system down to at least 50 mK. We note that at this early stage a concrete theoretical modeling of this strongly correlated phase is still lacking. It is a challenging task to compute the electronic structure of tWTe$_2$ moiré system even at the single particle level due to the large number of atoms and orbits involved in each moiré cell, together with the presence of spin-orbital coupling and possible superlattice reconstructions. Future theoretical consideration would also need to consider strong electron interactions, which seems to be essential in the system. The phenomenology of a 2D sliding LL, independently proposed two decades ago based on theoretical analysis of coupled-wire models[7–11], is well consistent with our observations here, including the large transport anisotropy, power law conductance across the wires and the vanishing differential resistance along the wires. However, establishing exact connections between the experiments and the models requires future efforts from both theory and experimental sides. Our experiments open new possibilities to further study topics related to 2D LL ground states and phase transitions, including spin-charge separation[1,20,25], novel quantum oscillations and quantum Hall effects in non-Fermi liquids[26–28].

## Methods

### Sample fabrication

We used Pd metal bottom and top gates in both devices. To create the metal bottom gate, a ~2 nm Ti/~6 nm Pd film was deposited onto an insulating Si/SiO$_2$ substrate using the standard e-beam lithography (EBL) and metal deposition tools. The bottom hBN was then transferred onto the metal bottom gate in a dry-transfer setup, followed by EBL and metal deposition of the metal contacts (~2 nm Ti/~6 nm Pd for device 1 and ~2 nm Ti/~6 nm Au for device 2). After a tip-cleaning process using an atomic force microscope (AFM), a thin hBN was transferred to cover the metal electrodes and selective areas of the thin hBN were etched using EBL and reactive ion etching (RIE) techniques. The top metal gate consists of either a ~15 nm Pd layer (device 1) or a ~3 nm Ti/~12 nm Pd layer (device 2), which was deposited onto exfoliated hBN flakes with the help of EBL for defining the location and shape. Monolayer WTe$_2$ was exfoliated on Si/SiO$_2$ substrates in an Ar-filled glovebox. The top metal/hBN stack was picked up as a whole followed by the 'tear-and-stack'[29,30] procedures that create the tWTe$_2$ stack. All processes that involve WTe$_2$ were conducted in the glovebox. A cartoon illustration of the fabrication process can be found in Supplementary Fig. 1.

### Transport measurement at ultralow temperatures

Electronic transport measurements were performed in a Bluefors dilution refrigerator with bottom loading probe system. The probe base temperature is ~24 mK. Thermocoax wires and a series of heat sinks are used to keep the electron temperature low. We calibrated the electron temperatures using a high-quality GaAs quantum well sample based on the activation behavior of a fractional quantum Hall state and found that the electron temperature in our setup starts to deviate from the probe thermometer temperature only below 45 mK. We therefore perform measurements in this work at temperatures of 50 mK or above.

In the measurements, an a.c. excitation of typically 10 μV with a frequency of 7 - 17 Hz, together with a d.c. bias, was applied to the source electrode via a Keysight 33511B function generator. Current and voltage signals were collected using lock-in amplifiers after a current (DL Instruments model 1211, with an internal impedance of 20 Ω) and voltage pre-amplifier (DL Instruments model 1201, with an internal impedance of 100 MΩ) to improve the signal. Gate voltages were applied via Keithley 2400 or 2450. The setup can reliably measure four-probe resistance up to a few MΩ and all data presented in this work are in the reliable regime.

### Data availability

The data that support the findings of this study are available at Harvard Dataverse (https://doi.org/10.7910/DVN/FVYOPF) or from the corresponding author upon request.

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

## Acknowledgements

We acknowledge discussions with Y.H. Kwan, S. A. Parameswaran, and S. L. Sondhi, S. A. Kivelson, and E. H. Fradkin. This work was supported by ONR through a Young Investigator Award (N00014-21-1-2804) to S.W. Measurement systems and data collection were supported by NSF through a CAREER award (DMR-1942942) to S.W. Materials synthesis and device fabrication were partially supported by the Materials Research Science and Engineering Center (MRSEC) program of the NSF (DMR-2011750) through support to R.J.C., L.M.S., and S.W. S.W. and L.M.S. acknowledge support from the Eric and Wendy Schmidt Transformative Technology Fund at Princeton. A.J.U. acknowledges support from the Rothschild Foundation and the Zuckerman Foundation. K.W. and T.T. acknowledge support from the JSPS KAKENHI (Grant Numbers 19H05790, 20H00354, and 21H05233). L.M.S. acknowledges support from the Gordon and Betty Moore Foundation through Grants GBMF9064, the David and Lucile Packard Foundation and the Sloan Foundation.

## Author contributions

G.Y. and P.W. fabricated the devices, performed measurements, and analyzed the data, assisted by A.J.U., Y.J., M.O., T.S., and Y.T., and supervised by S.W. R.S., L.M.S., X.G., and R.J.C. grew and characterized bulk WTe$_2$ crystals. K.W. and T.T. provided hBN crystals. S.W., G.Y., and P.W. wrote the paper with input from all authors.

## Competing interests

The authors declare no competing interests.
