## [Peer Review File · Nature Communications]

REVIEWER COMMENTS

Reviewer #1 (Remarks to the Author):

Wang et al. reported enhanced electronic anisotropy and power-law scaling behaviors in the conductance of twisted WTe₂. They argue that the system is a two-dimensional Luttinger liquid at the millikelvin temperature. The topic is exciting and the experiment provides a novel route to extend Luttinger liquid physics to higher dimensions. However, some questions need to be clarified before I can recommend the publication of this manuscript.

1. I found the results largely overlapped with the authors' previous paper published in Nature (Ref. 14). Although the authors argued that the current work explores a new temperature regime tuned by the twist angle, little is known about the behaviors of their previous device (theta ~ 5 degrees) at millikelvin temperatures. I'm also curious whether the authors have checked the transport behaviors of the untwisted WTe₂ or monolayer WTe₂ to firmly show that the effects are from the twist angle. Actually, in my opinion, to avoid simple duplication of the experiments, a twist-angle-dependent (in a relatively large range) experiment at millikelvin temperatures will improve the quality of the paper.

2. The claim of a two-dimensional Luttinger liquid in the title seems to be inappropriate. A 2D Luttinger liquid, in principle, should be isotropic. The system is more like a parallel 1D wires or quasi-1D system. The title used in the authors' previous paper in nature is more suitable.

3. I do not fully understand the different behaviors of the devices with 3 and 5 degrees twist angles. According to the authors' argument, the intra-wire interaction $\eta = (\alpha + 1)/2$. In both devices, alpha is large than 1, giving a value of eta larger than 1. Therefore, the crossover temperature should also vanish and the results reported in the current manuscript are also expected in the 5-degree device. It is also not clear to me why the LL behavior disappears above 2 K in the 3-degree device while it can persist up to 25 K in 5-degree device.

4. There may be a certain level of confusion in the current manuscript. In a system that is not strictly one-dimensional, it is well-known that there is a dimensional crossover energy E^* related to the perpendicular interaction (t_{\perp}). The system will exhibit a dimensional crossover between a 1D behavior at high energies and a more conventional high-dimension behavior at low energies. Theoretically, a LL would occur only if the dimensional crossover energy scale is strongly renormalized. The problem is, the energy scale in the current work is below 1 meV. Can the LL survive the dimensional crossover? Can the authors justify that the dimensional crossover energy scale is below the experimental energy scale?

If the authors can properly address the above questions, I will be happy to recommend the publication of the manuscript.

Reviewer #2 (Remarks to the Author):

The paper reports the observation of anisotropic 2D Luttinger Liquid (LL) behavior down to 50mK. This new electronic phase of matter, as first described by the authors in a 2022 paper in Nature, may yield novel correlated and topological electronic behaviors. The previous paper demonstrated the existence of 2D behavior in twisted WTe₂ layers (that act as coupled quantum wires) with twist angles of 5 and 6 degrees. This paper showed large (~1000:1) resistance anisotropies that developed only at low temperatures, and scaling behavior consistent with a 2D LL. That paper showed data at temperatures down to 1.8K. The cross-wire resistance the 5 and 6 degree twist angle samples (from the previous paper in Nature) grew too large to measure (beyond 10 megaohms) below 1.8K.

The current paper extends measurements to lower temperatures and also moves to lower twist angles of around 3 degrees (with interfere distance around 12 nm). The authors contend that this lower twist angle weakens the hopping interaction between wires and permits observation of LL effects to arbitrarily low temperatures. The authors claim, in their analysis of the data, to have achieved a value of the interaction parameter (η) that is greater than one permitting LL effects.

The current paper displays dramatic effects, with a resistance anisotropy greater than 10,000 and a superconductor-like dV/dI along the easy-axis direction at 50mK (figure 2h). The appropriate LL scaling (eV/kT) works down to the lowest temperatures and, by examining different regions of density and displacement field, the authors can find the conditions (apparently with a strong carrier density dependence but weak displacement field dependence) and produce a phase diagram for the 2D LL state.

Overall, I am excited about this work and believe that it should be published in Nature Communications. It is a clear and significant advance beyond the prior paper that was published in Nature. I am particularly interested in seeing more examination of the kind of data shown in figure 2h. In this regard, it would be nice to see if the superconductor-like dV/dI behavior in the easy-axis direction persists in the prior (5 and 6 degree twisted) samples. I know that WTe2 samples do not tend to survive well in storage, but if the authors have these data, it would be nice to include these in the extended data. I am curious about the effect of the interfere interaction on the existence of this superconductor-like state.

Starting on line 41 of page 3, the authors describe the small dV/dI seen in Figure 2h as a consequence of the easy-axis wires becoming disconnected from each other. This brings to mind the question of why there would be zero voltage along parallel wires. Is there something about the low temperature LL state that makes the wires behave independently? How much can we trust the zero voltage measurement - how can the wires be truly disconnected? What is the input impedance of the lock-in? It would be nice to have some discussion of this in the methods or elsewhere.

Note the typo - says T^a instead of T^α on page 4, line 21.

Reviewer #3 (Remarks to the Author):

In this paper, Yu et al. have demonstrated an experimental realization of Luttinger liquid (LL) in twisted WTe2 structures down to 50 mK. Due to its orthorhombic Td phase, WTe2 has a rectangular lattice which when twisted can be visualized as a system of 1D chains with a spatial separation determined by the twist angle. LL physics is strictly a 1D phenomenon which, due to the nature of the structure at hand, can be observed in this setup.

A few comments on the manuscript:

1. The authors use a dual-gate geometry in their devices which would allow them independent control of carrier density and electric field. Do the authors see any change in LL behaviour with displacement field for the same charge density?
2. In Fig. 4, scaling behaviour for point j in 4a too should be presented like has been done in 4h
3. Figure 1's caption (Page 10, line 8) erroneously refers number 12 whereas it should be 14

The experiment, although thoroughly conducted and analyzed, fails to present further new insights into LL physics or twisted WTe2 which the group's previous publication didn't

present (Ref. 14: Nature 605.7908 (2022): 57-62). The analysis is exactly the same: characterizing anisotropy, followed by scaling behaviour, and finally gate-dependence. The current manuscript does not bring enough novelty and consequently, I do not recommend its publication in Nature Communications.

Response to Referee Reports (manuscript # NCOMMS-23-09562-T)

We thank all referees for their constructive and insightful comments, which have motivated us to improve our manuscript. In the revised manuscript, we have now revised the discussions and main Figs. 1 & 4 and added 5 new Extended Data Figures, in response to the referees' questions. Changes in the manuscript are highlighted in yellow. Below we address the referees' comments one by one.

Reviewer #1 (Remarks to the Author):

Wang et al. reported enhanced electronic anisotropy and power-law scaling behaviors in the conductance of twisted WTe₂. They argue that the system is a two-dimensional Luttinger liquid at the millikelvin temperature. The topic is exciting and the experiment provides a novel route to extend Luttinger liquid physics to higher dimensions. However, some questions need to be clarified before I can recommend the publication of this manuscript.

Response: We thank the referee for the crisp summary of our results and valuable comments. We address the questions in the following.

1. I found the results largely overlapped with the authors' previous paper published in Nature (Ref. 14). Although the authors argued that the current work explores a new temperature regime tuned by the twist angle, little is known about the behaviors of their previous device ($\theta \sim 5$ degrees) at millikelvin temperatures. I'm also curious whether the authors have checked the transport behaviors of the untwisted WTe₂ or monolayer WTe₂ to firmly show that the effects are from the twist angle. Actually, in my opinion, to avoid simple duplication of the experiments, a twist-angle-dependent (in a relatively large range) experiment at millikelvin temperatures will improve the quality of the paper.

Response: We appreciate the referee's suggestions on including data on additional devices and agree that they will improve our manuscript. In the revised manuscript, we have now presented new data, including (1) Extended Data Fig. 2 for contrast between devices of monolayer WTe₂, 5° tWTe₂ and 3° tWTe₂; (2) Extended Data Fig. 3 for a 5° tWTe₂ device approaching millikelvin temperatures; and (3) Extended Data Fig. 4 showing behaviors of tWTe₂ with a range of twist angles from 2.7° to 10°.

In summary, we find that twist angle plays a key role in modifying physical properties of WTe₂ systems. For instance, near charge neutrality and on the hole side (the regime where we focus on the search for Luttinger liquid (LL) physics), monolayer WTe₂ is most insulating, reaching $\sim M\Omega$ at ~ 70 K near charge neutrality (Extended Data Fig. 2). Bilayer WTe₂ is typically less insulating compared to monolayer due to enhanced screening. At charge neutrality, 5° tWTe₂ can be reliably measured down to ~ 2 K, whereas the resistance quickly becomes larger than $\sim M\Omega$ when it falls in millikelvin regime (Extended Data Fig. 3, also previously shown in our Nature paper). Note that the standard lock-in measurement technique works reliably for up to $\sim M\Omega$ in four-probe measurements due to the internal resistance of voltage pre-amplifiers, a known issue in transport measurements. A two-probe measurement can probe resistance up to larger values, but it inevitably contains contact resistance especially if the sample is insulating. 3° tWTe₂ is further less insulating than 5° tWTe₂ and the resistance reaches $\sim M\Omega$ at ~ 50 mK, thus enabling reliable, quantitative experiments to evaluate its transport behavior at millikelvin. This trend of device resistance is summarized in Extended Data Figs. 2 & 4.

From these data, we conclude the following. (1) The twist angle plays a crucial role in modifying the properties of WTe₂. LL transport characteristics are so far only observed in tWTe₂ (not in the monolayer or natural bilayer samples). (2) While LL physics can be consistently observed in a range of small twist angles (i.e., there is no magic angle), the temperature window for the LL to occur varies.

(3) We cannot experimentally conclude on the exact behaviors of the LL state in the 5° tWTe₂ device at millikelvin due to challenges in measuring large resistances (a crossover to a non-LL state is generally expected for most doping, as seen on the hole side of the 3° tWTe₂ device; also please refer to our response below to question 3); (4) For the purpose of examining the possible existence of a LL state down to ~ 50 mK using transport measurement, which requires the hard-direction resistance not larger than $\sim M\Omega$ at such low T , the only choice we found suitable is a twist angle of near $\sim 3^\circ$. This is hence the focus of the current manuscript.

2. The claim of a two-dimensional Luttinger liquid in the title seems to be inappropriate. A 2D Luttinger liquid, in principle, should be isotropic. The system is more like a parallel 1D wires or quasi-1D system. The title used in the authors' previous paper in nature is more suitable.

Response: We agree with the referee's concern. In our original title, we intended to highlight that the phase is of a 2D system but agree that its anisotropy should also be highlighted. We have changed the title to “*Evidence for Two Dimensional Anisotropic Luttinger Liquids at Millikelvin Temperatures*”.

3. I do not fully understand the different behaviors of the devices with 3 and 5 degrees twist angles. According to the authors' argument, the intra-wire interaction $\eta = (\alpha + 1)/2$. In both devices, α is large than 1, giving a value of η larger than 1. Therefore, the crossover temperature should also vanish and the results reported in the current manuscript are also expected in the 5-degree device. It is also not clear to me why the LL behavior disappears above 2 K in the 3-degree device while it can persist up to 25 K in 5-degree device.

Response: The referee's question is indeed crucial in understanding the experimental data. We first clarify our discussion regarding the theoretical background. The simplified phase diagram of coupled wires, as a function of the effective intrawire interaction η , features a regime of $\eta > 1$, where *single-particle* interwire hopping (t_\perp) becomes irrelevant (Fig. 1b). This is hence the regime where we search for a LL state in 2D at vanishing temperatures. However, this discussion only considers *single-particle* hopping process. In realistic systems, other competing phases, e.g., those arising from *two-particle* hopping processes, become important, providing instabilities to the LL state especially in this regime (see e.g., Fig. 2 in *PRL* **85**, 2160–2163 (2000) for possible competing phases). Evaluating the competing phases and LL instabilities is indeed the key focus of multiple theoretical works in early 2000s, e.g., Wen *PRB* **42**, 6623–6630 (1990); Emery, Fradkin, Kivelson & Lubensky *PRL* **85**, 2160–2163 (2000); Sondhi & Yang *PRB* **63**, 054430 (2001); Vishwanath & Carpentier *PRL* **86**, 676–679 (2001) & Mukhopadhyay, Kane, & Lubensky *PRB* **64**, 045120 (2001) (cited as refs 7-11). In this regime, a single intrawire interaction is no longer sufficient for fully characterizing the physics. Instead, one needs to consider both interwire and intrawire interactions with momentum dependence, a much more complex situation.

In other words, a large η (approaching to or larger than 1) signifies a new regime where (i) single-particle interwire hopping becomes irrelevant and (ii) consequently the simplified description based on a single interaction parameter η and only single-particle hopping breaks down. A large η doesn't guarantee a LL state down to zero temperature in the presence of two-particle hopping terms and competing instabilities; indeed, most of the parameter space of the calculated phase diagram features other competing orders instead of a LL (see e.g., *PRL* **86**, 676–679 (2001)). The central theoretical question is whether there is any finite parameter space where *All* interwire hopping processes become irrelevant at vanishing temperatures, a condition for a LL state to truly survive as a ground state. The conclusion in these theories is that, while competing orders reside in most regions of the phase

diagram, it is luckily possible that, in some small, fine-tuned parameter space, there in principle exists an anisotropic 2D LL state, dubbed a “sliding LL” (see e.g., *PRL* **86**, 676–679 (2001)). Experimentally it is challenging to fine tune interactions in conventional materials and hence an experimental test of this fascinating idea was lacking.

From this perspective, our works on $tWTe_2$ identified a highly tunable 2D moiré system for exploring these possibilities. The central question is now that, experimentally, whether one can observe evidence of a 2D anisotropic LL phase at vanishing temperatures.

Therefore, although in the 5° device we previously found cases of α above 1 in the measurement above 1.8 K, we do expect theoretically that the LL state may crossover to a different phase at mK in the presence of two-particle hopping process. This crossover is indeed seen in the hole side of the 3° device (Fig. 4j). Experiments in the millikelvin region are necessary to determine whether an LL phase in any 2D system can survive as a stable ground state in a finite region of the phase diagram. A careful transport characterization requires a quantitative measurement of the resistance. As we have explained above (and in Extended Data Fig. 3), the resistance of the 5° device quickly becomes too large ($> M\Omega$) at millikelvin, preventing a reliable evaluation. This is the reason why we chose to study $tWTe_2$ devices with a smaller angle of $\sim 3^\circ$ as they allow us to analyze transport characteristics at mK quantitatively.

We further note that a precise theoretical description of $tWTe_2$ electronic structure, and subsequently a concrete understanding of connections between our experiments and the theory of sliding LL, remains challenging. The experiments do show a consistent trend for the appearance of LL at small twist angles, with the LL temperature window shifting down to a lower region when the angle is reduced from 5° to 3° . The emergence of LL state starts at ~ 25 K for the 5° device while at ~ 2 K for the 3° device, which reflects the modified hopping and interaction parameters in the system. Note that the moiré unit cell of a $3^\circ tWTe_2$ is larger than that of a 5° device, i.e., the “wires” are expected to be wider and of a larger separation between wires, consistent with a reduced energy scale. However, even at the density functional theory level, the properties of $tWTe_2$ are difficult to compute due to many orbitals in one unit-cell, strong spin-orbit coupling, and non-trivial interaction effects. At this stage, we cannot predict the exact temperature range for the LL phase to appear at a given twist angle or gate voltage, which requires new theoretical developments on complex moiré band structures, interactions and competing orders. Our search for an LL phase in $tWTe_2$ here relies on experimental observations.

We thank the referee for bringing up this important point. We have revised our discussions to include the above discussions more explicitly.

4. There may be a certain level of confusion in the current manuscript. In a system that is not strictly one-dimensional, it is well-known that there is a dimensional crossover energy E^* related to the perpendicular interaction (t_{perp}). The system will exhibit a dimensional crossover between a 1D behavior at high energies and a more conventional high-dimension behavior at low energies. Theoretically, a LL would occur only if the dimensional crossover energy scale is strongly renormalized. The problem is, the energy scale in the current work is below 1 meV. Can the LL survive the dimensional crossover? Can the authors justify that the dimensional crossover energy scale is below the experimental energy scale?

Response: We thank the referee for the essential questions. Our hope in this work is indeed to address them, by pushing the measurement down to 50 mK ($\sim 5 \mu\text{eV}$!). This is well below the energy scale (\sim

meV) of the hopping and interaction terms in the system. Experimentally, in the $\sim 3^\circ$ device our finding of the LL characteristics down to at least 50 mK implies that any dimensional crossover, if exists, must be lower than this temperature. Thus, the experiment has set an upper bound of $\sim 5 \mu\text{eV}$ for the energy scale of any dimensional crossover, well below the interaction energy scale ($\sim \text{meV}$) in the system. We further comment on several aspects.

- (1) Regarding dimensional crossover - The dimensional crossover energy, if only considering single particle hopping process, will be renormalized as $E^* \sim t_{\perp}(t_{\perp}/\hbar\eta)^{\eta/(1-\eta)}$, where t_{\perp} ($\hbar\eta$) is the interwire (intrawire) hopping term and η is the power law exponent reflecting intrawire interaction. If η approaches to or is larger than 1, this dimensional crossover energy will be fully suppressed to zero. However, when one considers two-particle processes, as discussed in the response above, other competing orders become important, and the theoretical description is in a much more complex situation (see refs. 7-11 for theoretical considerations). Our approach is to directly search for a LL state in 2D experimentally at vanishing temperatures: we push the temperature to the lowest, first down to $\sim 1.8 \text{ K}$ in our nature paper [Nature 605, 57–62 (2022)], and now down to 50 mK in this work.
- (2) At millikelvin temperatures, we do find a large gate range on the hole side where the LL state crossovers to a non-LL state (see Fig. 4j for a preliminary phase diagram). The region where LL state survives down to 50 mK locates in a relatively narrow regime near charge neutrality. This is consistent with the presence of competing orders.
- (3) The demonstration of the existence of a 2D anisotropic LL down to such a low temperature (50 mK) is an important advance for not only setting an exceptionally low upper bound of any dimensional crossover energy, but also establishing a realistic platform for examining new physics based on coupled-LL arrays. An example is the possible realization of integer and fractional quantum Hall effects in such a non-Fermi liquid [e.g., PRL 88, 036401 (2002)], which are expected to occur at low temperatures (likely in the millikelvin regime). A realization of the LL state down to this temperature range sets a stage for these future explorations.

We agree with the referee that in our original manuscript there might be some level of confusion in the discussion of dimensional crossover, especially the distinguishment between analysis based on single-particle process and the more sophisticated theory considerations (refs. 7-11). In the revised manuscript, we have revised related discussions and avoided potential confusions in the main text. We have also revised Fig. 1b, where the word “2D Luttinger liquid” is now changed to “possible 2D anisotropic LL at vanishing temperature?”.

If the authors can properly address the above questions, I will be happy to recommend the publication of the manuscript.

Response: We hope we have addressed the referee’s questions above. We are grateful for the referee’s insightful and constructive comments, which have substantially improved our manuscript.

Reviewer #2 (Remarks to the Author):

The paper reports the observation of anisotropic 2D Luttinger Liquid (LL) behavior down to 50mK. This new electronic phase of matter, as first described by the authors in a 2022 paper in Nature, may yield novel correlated and topological electronic behaviors. The previous paper demonstrated the existence of 2D behavior in twisted WTe2 layers (that act as coupled quantum wires) with twist angles of 5 and 6 degrees. This paper showed large ($\sim 1000:1$) resistance anisotropies that developed only at low temperatures, and scaling behavior consistent with a 2D LL. That paper showed data at

temperatures down to 1.8K. The cross-wire resistance the 5 and 6 degree twist angle samples (from the previous paper in Nature) grew too large to measure (beyond 10 megaohms) below 1.8K.

The current paper extends measurements to lower temperatures and also moves to lower twist angles of around 3 degrees (with interfere distance around 12 nm). The authors contend that this lower twist angle weakens the hopping interaction between wires and permits observation of LL effects to arbitrarily low temperatures. The authors claim, in their analysis of the data, to have achieved a value of the interaction parameter (η) that is greater than one permitting LL effects.

The current paper displays dramatic effects, with a resistance anisotropy greater than 10,000 and a superconductor-like dV/dI along the easy-axis direction at 50mK (figure 2h). The appropriate LL scaling (eV/kT) works down to the lowest temperatures and, by examining different regions of density and displacement field, the authors can find the conditions (apparently with a strong carrier density dependence but weak displacement field dependence) and produce a phase diagram for the 2D LL state.

Overall, I am excited about this work and believe that it should be published in Nature Communications. It is a clear and significant advance beyond the prior paper that was published in Nature. I am particularly interested in seeing more examination of the kind of data shown in figure 2h. In this regard, it would be nice to see if the superconductor-like dV/dI behavior in the easy-axis direction persists in the prior (5 and 6 degree twisted) samples. I know that WTe₂ samples do not tend to survive well in storage, but if the authors have these data, it would be nice to include these in the extended data. I am curious about the effect of the interfere interaction on the existence of this superconductor-like state.

Response: We thank the referee for the thorough and insightful summary of our results and are grateful for their support in the publication of this work.

Indeed, the referee's comment on the nonlinear "superconductor-like" dV/dI feature is an important new finding in this work for characterizing the 2D anisotropic LL phase. We agree with the referee that it will be interesting to examine similar effects in our previous device on $\sim 5^\circ$ tWTe₂. Following the referee's suggestion, we present the data now as Extended Data Fig. 7 in the revised manuscript. As shown, we have observed consistent nonlinear features in the easy-axis dV/dI data. We further note two differences compared to the 3° tWTe₂ device. First, the zero-bias dip of dV/dI is much wider in the 5° device, indicating a larger energy scale associated with the LL state in the 5° device. This is consistent with the fact that its LL characteristics occur at higher temperatures (the LL feature starts to develop already at ~ 30 K in the 5° device whereas it is below ~ 1 K in the 3° device). Second, the zero-bias dip in the 5° device doesn't reach zero (albeit a clear dip) at the temperature of 1.8 K. The dip may further develop if the temperature is lowered to millikelvin, a regime where a reliable measurement is difficult due to large sample/contact resistance. Overall, the observation of the nonlinear dV/dI feature is consistent between devices, serving as an additional key characteristic of the 2D anisotropic LL phase. This aspect is relevant to the referee's next question we address below.

Starting on line 41 of page 3, the authors describe the small dV/dI seen in Figure 2h as a consequence of the easy-axis wires becoming disconnected from each other. This brings to mind the question of why there would be zero voltage along parallel wires. Is there something about the low temperature LL state that makes the wires behave independently? How much can we trust the zero voltage

measurement - how can the wires be truly disconnected? What is the input impedance of the lock-in? It would be nice to have some discussion of this in the methods or elsewhere.

Response: We first assure that the nonlinear dV/dI (i.e., the zero-bias dip) is real and the vanishing voltage is measured reliably within experimental uncertainties. The input impedance of the instrument directly connected to the device is $100\text{ M}\Omega$ (from the voltage preamplifier DL Instruments model 1201). We can hence reliably measure four-probe resistance up to $\sim\text{M}\Omega$. All data presented in this work are in the regime where the resistance can be reliably determined. We also note that the zero-bias dip gradually develops starting from a relatively high temperature or bias, assuring that the feature of vanishing dV/dI at zero bias is reliable.

The referee’s question on how the wires can behave independently, as indicated by the vanishing zero-bias dV/dI , is insightful. We indeed attribute this observation as one of the remarkable features of the new 2D anisotropic LL phase.

In the analysis of coupled-wire models, the predicted 2D “sliding LL” is indeed a strongly anisotropic phase that mimics an array of seemingly “independent” wires (see refs. 7-11). This appears not due to an absence of interwire interactions; instead, it describes a highly nontrivial strongly interacting phase where correlations stabilize a situation in which *all interwire hopping processes become irrelevant*. For instance, in the theory work of Vishwanath & Carpentier *PRL*, **86**, 676–679 (2001), transport in the transverse direction (across-wire) is described as a “perfect insulator” (at zero temperature). The experimental consequence is that, at vanishing temperature, the system behaves as a LL in each wire while transport across wires is fully suppressed (unless at a finite temperature or bias). We believe the observed zero-bias dip in the easy-axis, together with the large anisotropy and the power law conductance in the hard-axis, indicates the emergence of a highly intriguing new phase akin to the proposed “sliding LL”, although the exact understanding of the connections requires substantial future developments in both theory and experiments.

We note that, different from a bundle of nanowires, in the $t\text{WTe}_2$ system we do not start with individual wires (the $3^\circ t\text{WTe}_2$ system is quite isotropic even just at about 2 K). The aspect ratio of the rectangular $t\text{WTe}_2$ moiré cell is about 1:2, which is anisotropic but cannot fully justify the measured exceptionally large resistance anisotropy of more than 10,000. Hence electron interactions play a key role to spontaneously stabilize the highly anisotropic phase that mimics an array of independent LL wires at millikelvin temperatures. The nonlinear dV/dI (Fig. 2h) indicates that this process occurs like a superconducting transition at the Fermi surface; the difference is that here it transits to a 2D LL state, instead of a superconducting state.

We thank the referee for pointing out this important aspect. In the revised manuscript, we have included the instrument internal resistance in the Method section and highlighted the implications of the zero bias dip further in the main text.

Note the typo - says T^a instead of T^α on page 4, line 21.

Response: We have corrected the typo. We thank the referee for pointing this out.

Reviewer #3 (Remarks to the Author):

In this paper, Yu et al. have demonstrated an experimental realization of Luttinger liquid (LL) in twisted WTe_2 structures down to 50 mK. Due to its orthorhombic T_d phase, WTe_2 has a rectangular

lattice which when twisted can be visualized as a system of 1D chains with a spatial separation determined by the twist angle. LL physics is strictly a 1D phenomenon which, due to the nature of the structure at hand, can be observed in this setup.

Response: We thank the referee for the concise summary of our work. Below we address the comments.

A few comments on the manuscript:

1. The authors use a dual-gate geometry in their devices which would allow them independent control of carrier density and electric field. Do the authors see any change in LL behavior with displacement field for the same charge density?

Response: Yes, we have indeed explored the effect of the electric displacement field D . We have now included a new Extended Data Fig. 10 to summarize the observation. We find that while LL behaviors are sensitive to carrier density, the displacement field effect is less dramatic especially if D is not too large. At high D , we do find a drop in the power law exponent (Extended Data Fig. 10d), potentially indicating a transition to a different phase if D is further increased. We have also included a discussion on the displacement field effect in the revised main text.

2. In Fig. 4, scaling behavior for point j in 4a too should be presented like has been done in 4h

Response: We thank the referee for the suggestion and have revised Fig. 4 to include the data.

3. Figure 1's caption (Page 10, line 8) erroneously refers number 12 whereas it should be 14

Response: We thank the referee for pointing out this typo, it has now been corrected.

The experiment, although thoroughly conducted and analyzed, fails to present further new insights into LL physics or twisted WTe₂ which the group's previous publication didn't present (Ref. 14: Nature 605.7908 (2022): 57-62). The analysis is exactly the same: characterizing anisotropy, followed by scaling behavior, and finally gate-dependence. The current manuscript does not bring enough novelty and consequently, I do not recommend its publication in Nature Communications.

Response: We thank the referee for the comment. While it is true that here we use the same transport approach to characterize the LL state, there are key differences in this work that are highly nontrivial and represent an essential step forward for advancing the concept of a stable LL in 2D. Below we highlight the novelty of this work compared to our previous work in Nature.

(1) *New characteristics of the 2D anisotropic LL state (nonlinear dV/dI in the easy-axis)* - As highlighted by referee 2, in this work, in addition to the transport anisotropy and power law conductance in the hard direction we report for the first time the nonlinear dV/dI features (zero bias dip) in the four-probe easy-axis measurements (Fig. 2h and Extended Data Fig. 6b & d). The vanishing dV/dI in the zero-bias dip provides another key characterization of the new 2D anisotropic LL phase, not presented in our previous nature paper. The dV/dI data signifies a transition at the Fermi surface upon lowering the temperature or bias: the system transits to a 2D non-Fermi liquid phase where each wire behaves as if there were "independent" LLs. In the predicted 2D "sliding LL" phase of coupled-wire models (see Vishwanath & Carpentier *PRL*. **86**, 676–679 (2001), or refs. 7-11), correlations stabilize a situation

in which *all interwire hopping processes become irrelevant*. The dramatic zero-bias dip hence provides a key new characterization of the emergence of an intriguing new 2D phase akin to the “sliding LL”. Although the understanding of exact connections between the experiment and the theories requires substantial future developments, our results in this work, including the addition of the easy-axis zero-bias dip in the dV/dI , provide an experimental platform for such explorations.

(2) *A 2D anisotropic LL down to 50 mK* – This is our main finding in this work. We first highlight that whether a LL phase can be a stable ground state in 2D is a highly nontrivial question even in theory. In general, the LL phase in a 2D array of coupled wires, is only expected to exist in a finite temperature range, below which a dimensional crossover to a non-LL phase occurs. Evaluating competing phases at low temperatures in the coupled-wire constructions was the key focus of multiple theoretical works, e.g., refs 7-11. The conclusion in these theories is that, while competing orders other than a LL reside in most of the phase diagram, it is possible that in some fine-tuned parameter space there exists an anisotropic LL state in 2D at vanishing temperatures (see e.g., *PRL* **86**, 676–679 (2001)). In conventional materials, it is extremely challenging to fine-tune interactions and hence experimental tests of the concept were lacking. Our works on $tWTe_2$ identify a highly tunable 2D moiré system for exploring these possibilities. The key to advance the topic is to experimentally search for a 2D LL state in a real material at vanishing temperatures.

From this perspective, we believe a key novelty of this current work is that we have indeed found a 2D anisotropic LL phase surviving down to at least 50 mK, which sets an exceptionally low upper bound for any dimensional crossover energy (more than one order of magnitude lower than the previous record of 1.8 K observed in our Nature paper). This represents one crucial step further in the search for a stable LL beyond 1D.

The demonstration of the existence of a 2D anisotropic LL down to such a low temperature (~ 50 mK) also establishes a practical platform for examining new physics based on coupled-LL arrays. An example is the possible observation of integer and fractional quantum Hall effects in such a non-Fermi liquid [see e.g., *PRL* **88**, 036401 (2002)]. These phenomena are expected to occur only at low temperatures (likely in the millikelvin regime). A demonstration of the LL state in this temperature regime sets a stage for exciting explorations in the future. We hope we have convinced the referee that our work has indeed demonstrated new characterizations of the LL state and extended the physics to a new regime unexplored in our previous work.

REVIEWERS' COMMENTS

Reviewer #1 (Remarks to the Author):

The authors properly answered most of my questions. However, I am still concerned about the claim of a "2D LL". On the one hand, the LL phase in the current work is due to the "hidden" 1D structure similar to the previous work in MoS₂ with domain walls. On the other hand, the mechanism of why a LL can exist in 2D is not sufficiently explained. If this point can be properly addressed, I will be happy to recommend the publication of the paper.

Reviewer #2 (Remarks to the Author):

I am satisfied with the authors' responses to my questions and believe that the paper is suitable for publication in Nature Communications without further delay.

Reviewer #3 (Remarks to the Author):

The authors have satisfactorily answered my concerns and added additional data to the manuscript supporting their claims. I recommend the publication of this paper in Nature Communications.

Response to Referee Reports (manuscript # NCOMMS-23-09562A)

We are grateful to all three referees for their efforts in reviewing our manuscript. Below we address the remaining point of referee #1.

Reviewer #1 (Remarks to the Author):

The authors properly answered most of my questions. However, I am still concerned about the claim of a "2D LL". On the one hand, the LL phase in the current work is due to the "hidden" 1D structure similar to the previous work in MoS₂ with domain walls. On the other hand, the mechanism of why a LL can exist in 2D is not sufficiently explained. If this point can be properly addressed, I will be happy to recommend the publication of the paper.

Response: We thank the referee for the additional question. We first clarify that the LL phase observed in our case is not due to presence of domain-wall-like 1D atomic structure. In the tWTe₂ system we do not start with individual 1D wires/domain walls (no "hidden" 1D atomic structures). The host of the 2D anisotropic LL state is the moiré lattice of tWTe₂, whose super unit cell is a rectangle with an aspect ratio of about 1:2. Namely, the moiré system is anisotropic but clearly still a 2D lattice. The small lattice anisotropy, however, cannot justify the measured exceptionally large electric transport anisotropy, $R_{\text{hard}}/R_{\text{easy}}$, of more than 10,000. Hence electron interactions must play a key role to the formation of this highly anisotropic (but 2D) electronic phase that mimics an array of LL wires at millikelvin temperatures. As we reported in this work, the experimental phenomena of this phase include (1) an exceptionally large transport anisotropy, (2) a power-law scaled conductance in the hard direction and (3) a nonlinear dV/dI that vanishes at zero bias in the easy direction. We are not aware of any other experimental work in the literatures (including MoS₂ literatures) that shows these three key features together. These are key characterizations of the new 2D electronic phase.

While we have emphasized that understanding the exact mechanism of this intriguing new phase in tWTe₂ requires future inputs especially from theory side, the general question of why a LL can exist 2D has been addressed in a series of theoretical works based on coupled wire models, e.g., Wen *PRB* **42**, 6623–6630 (1990); Emery, Fradkin, Kivelson & Lubensky *PRL* **85**, 2160–2163 (2000); Sondhi & Yang *PRB* **63**, 054430 (2001); Vishwanath & Carpentier *PRL* **86**, 676–679 (2001) & Mukhopadhyay, Kane, & Lubensky *PRB* **64**, 045120 (2001) (refs 7-11). These theoretical works have provided a concrete mechanism, owing to strong electron interactions, for the existence of a highly anisotropic 2D phase that mimics an array of independent 1D Luttinger liquids (the "sliding Luttinger liquid"). At this moment, we avoid claiming that we have realized the exactly same phase, but the experimental phenomena (i.e., the above three characteristics) are consistent with the expectations for this novel phase or its variations. Discussions regarding this theory aspect have been included in our manuscript. We hope we have now fully addressed the referee's questions.

Reviewer #2 (Remarks to the Author):

I am satisfied with the authors' responses to my questions and believe that the paper is suitable for publication in *Nature Communications* without further delay.

Response: We thank the referee for recommending the publication of our work.

Reviewer #3 (Remarks to the Author):

The authors have satisfactorily answered my concerns and added additional data to the manuscript supporting their claims. I recommend the publication of this paper in *Nature Communications*.

Response: We thank the referee for recommending the publication of our work.